# How Vegetation Colorization Design Affects Urban Forest Aesthetic Preference and Visual Attention: An Eye-Tracking Study

Ziru Chen [1], Yaling Huang [2], Yuanping Shen [1], Weicong Fu [2,3], Xiong Yao [1], Jingkai Huang [2], Yuxiang Lan [1], Zhipeng Zhu [1] and Jiaying Dong [4,*]

1   College of Architecture and Planning, Fujian University of Technology, Fuzhou 350108, China; fjchenziru@fjut.edu.cn (Z.C.); 19801066@fjut.edu.cn (Y.S.); 19902197@fjut.edu.cn (X.Y.); 19912325@fjut.edu.cn (Y.L.); 19912151@fjut.edu.cn (Z.Z.)
2   College of Landscape Architecture and Art, Fujian Agriculture and Forestry University, Fuzhou 350002, China; wwwhyling125@163.com (Y.H.); weicong.fu@fafu.edu.cn (W.F.); 13375002763@163.com (J.H.)
3   Collaborative for Advanced Landscape Planning, Faculty of Forestry, The University of British Columbia, Vancouver, BC V6T 1Z4, Canada
4   School of Architecture, Clemson University, Clemson, NC 29634, USA
*   Correspondence: jiayind@clemson.edu; Tel.: +86-182-5036-5086

**Abstract:** The enhancement of the urban forest landscape through vegetation colorization has emerged as a continuous concern for urban managers in southern Chinese cities. However, the understanding of how designers can effectively select the appropriate form and intensity of colorization design to align with users' aesthetic preferences remains limited. The process of visual perception is closely intertwined with eye movements. Employing visualization techniques, this research aims to investigate the impact of colorization design on aesthetic benefits and eye movements in urban forests, considering four intensities (slight, low, medium, and high) and three forms (aggregate, homogeneous, and random). A total of 183 participants (with an average age of 23.5 ± 2.5 years) were randomly assigned to three groups to assess the aesthetics score, while eye-tracking devices were utilized to record eye movement behaviors. The outcomes indicate that a homogeneous design form and a moderate intensity of landscaping yield higher scenic benefits for urban forests. In the case of canopy landscape, both the form and intensity of landscaping have a significant influence on urban forest aesthetics. The HCI with aggregate form showed the best marginal effect (1.313). In contrast, MCI showed the best marginal effect when the design form was random and homogeneous (1.438, 1.308). Furthermore, although the form and intensity of the colorization design significantly affect eye exploration, the perception of landscape aesthetics does not correlate with eye movements. These findings provide valuable insights for design policies aimed at promoting the urban forest landscape, while also contributing to the enrichment of research in landscape perception studies employing eye-tracking technology.

**Keywords:** landscape colorization; design form; design intensity; eye movement; aesthetic rating

## 1. Introduction

Enhancing the aesthetic quality of urban forest areas through design is a significant concern for urban managers and designers due to its pivotal role in urban composition. Previous research emphasized the substantial enhancement of landscape aesthetic value through the introduction of colored vegetation [1–3]. Similarly, Chen noted that urban residents express a preference for living in a landscape that exhibits distinct seasonal changes [4]. This preference is particularly pronounced in southern Chinese cities, where seasonal variations are relatively subtle. Furthermore, landscapes featuring vibrant vegetation have been found to possess a heightened appeal to individuals [5]. Kendal et al.

demonstrated that the presence of colored vegetation in green spaces perceptibly enhances the perception of ecological diversity within the landscape [6]. Zhang's research corroborated these findings by revealing that subjective ratings for blue, red, and pink flowers were comparatively high [7]. Moreover, studies have demonstrated the feasibility of vegetation colorization design for urban forests in southern China [8,9]. Notably, tree species such as Maple and Sapium sebiferum, which exhibit red foliage, as well as Cottonwood and Lambsquarters, which produce vibrant flowers, can thrive and create a colorful impact in the southern Chinese region [10].

In the realm of implementing urban forest spaces, prior research has indicated the utility of visualization in enabling designers to gain a precise understanding of design content [3,11–13]. Xu's study evaluated various design intensity scenarios across different forest landscape types, revealing a preference for landscapes featuring a moderate design intensity of artificial elements, as opposed to high or low design intensities [14]. Similarly, Dupont argued that landscapes with low saliency are more readily accepted [15]. However, these studies primarily examined the effects of high, medium, and low design intensities without quantifying the specific design intensities, thereby limiting their practical generalizability. Drawing upon the perceived capacity of the human body, Dehaene demonstrated that the proportion of elements and human perception adhere to the Weber–Fechner law. Supporting this theory, Jack et al. substantiated the influence of both the design intensity and form of elements on landscape effects [16]. Moreover, Sheppard posited that the perception of urban forest landscapes from within and outside the forest differs and necessitates separate consideration in the context of forest landscape creation [17]. Nevertheless, only a few studies have investigated urban forest green space construction from both in-forest and out-of-forest perspectives [18]. By examining urban forest green space enhancement in terms of design form and intensity from different vantage points, effective cost–benefit control can be achieved for enhancing urban forest landscapes. This research aims to quantitatively assess design intensity and explore the relationship between colorization design intensities and forms concerning landscape aesthetic preferences in the urban forest landscapes of southern China, specifically focusing on canopy landscapes and forest landscapes. The findings of this research will offer novel insights and guidelines for the design of urban forest spaces.

In recent decades, research on visual aesthetic quality has undergone significant development, resulting in the establishment of a relatively mature system. Previous studies primarily relied on questionnaires to collect expert or public evaluations [19–21]. These evaluations often involved a series of procedures such as visual perception and conscious judgment [22]. While medieval theory postulated the separation of introspection from conscious perception, foundational cognitive theories have enabled researchers to move beyond these elegant formalisms [23–25]. Vision serves as the most direct means through which humans perceive landscapes. By capturing eye-movement behavior, it becomes possible to gain intuitive insights into the brain's perceptual judgments [26–28]. Consequently, it is essential to explore the relationship between eye movements and landscape evaluation in research endeavors. A limited number of studies have unveiled the inherent patterns of eye movements and their connection to landscape perception in humans [29]. Existing research works have demonstrated the feasibility of employing eye-tracking technology in landscape perception studies [30]. They have revealed a certain degree of association between eye movements and landscape perception. However, these studies often drew their conclusions by comparing extensive settings, thereby making it challenging to apply their findings in practical design scenarios. Moreover, the impact of different design methods employed in urban forest landscape settings on eye movements, including factors such as pupil size, fixation, and saccades, remains poorly understood [31–33]. Therefore, there is a pressing need to expand our understanding of this association by incorporating eye-tracking technology into human perception research endeavors.

Previous research has established that different landscape features elicit distinct patterns of eye movements [32,34], which subsequently shape landscape perception [35]. Berto's study demonstrated notable differences in fixation patterns when individuals viewed landscapes of varying aesthetic value for aesthetic purposes [36]. Landscapes with lower aesthetic value were found to prompt more exploration and longer fixation durations compared to landscapes with a higher aesthetic value [37]. Additionally, visual exploration is influenced by the combination of landscape elements, with individuals directing more attention toward elements that pique their interest. This phenomenon is particularly prominent in the color representation of landscape elements [24,38]. Consequently, it can be inferred that the aesthetic value of a landscape stimulates increased fixation frequency and duration [39]. Furthermore, in the context of enhancing green space landscapes through colorization, landscape design should not only consider design intensity but also design form, given the specific scenario conditions. However, the literature lacks sufficient reports on how to achieve optimal design combinations within limited conditions to enhance the aesthetic appearance and effectively attract visual attention and exploration from subjects. Such research results could provide more precise enhancement strategies for the landscape enhancement of mountains as a way to improve the urban landscape.

Based on field research, the photomontage method was applied to visualize the scene of four intensities (slight, low, medium, and high) under three forms of colorization design (aggregate, homogenous, and random) in canopy landscape and forest landscape of Yushan Scenic Area. Eye-movement behaviors and landscape aesthetic values were recorded for exploring the relationship. The research aimed to address the following questions:

1. Which colorization design form and intensity is more effective in enhancing the aesthetic value of urban forest landscapes?
2. How do different forms and intensities of colorization design influence eye movements?
3. What is the relationship between the aesthetic value of the urban forest landscape and the eye-movement metrics?

By integrating the intensity and form of colorization design, this research aims to provide comprehensive design and management guidance for urban forest designers and decision makers. The incorporation of eye movement technology offers a deeper understanding of object-perceived landscape research utilizing eye-tracking methodologies.

## 2. Materials and Methods

### 2.1. Study Site

The research site Yushan Scenic Area is located in Fuzhou, China, which is the representative urban green space of Fuzhou city, and can be considered as the image of Fuzhou city (Figure 1). Fuzhou, located on the southeast coast of China, predominantly features mountainous urban forest areas characterized by evergreen vegetation that lacks noticeable seasonal variations. The research focused on the canopy landscape and nine selected recreational forest landscapes within the Yushan Scenic Area. Visualization simulations were conducted during the autumn season, as it is a period when residents anticipate and appreciate seasonal changes the most [4,40].

### 2.2. Photographic Images and Visualizations

Photographs have been extensively validated as authentic and effective substitutes for real-life scenes in numerous studies [4,22,41]. Therefore, this research employed photographs as stimuli. To capture the entire landscape of Yu Mountain, a UAV (unmanned aerial vehicle) was utilized to photograph the area from a 45° angle downward in October 2021, serving as the base scene for generating and comparing the canopy visualization scenes. Additionally, nine popular forest scenes were selected for the forest landscape component. A Nikon D900 camera was employed to capture images at a human-view height of 1.5 m [42].

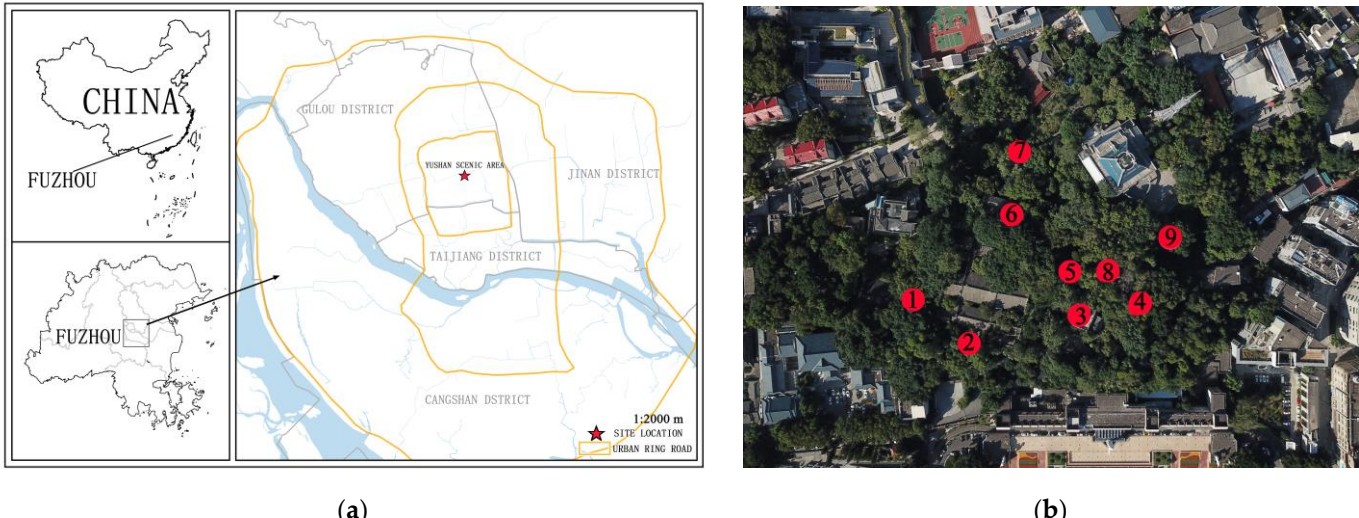

(**a**)　　　　　　　　　　　　　　　　　(**b**)

**Figure 1.** Study Site. (**a**) Fuzhou location. (**b**) Forest observation site. 1–9 in (**b**): 9 forest scenes of observation.

In order to determine the space available for planting plants in Yu Shan Park, and to provide practical guidance, the researcher conducted field surveys in the Yushan scenic area with an S760 handheld centimeter GPS data collector. Potential sites for vegetation planting design were identified by marking specific locations that already exhibited autumn colors in the plant display and had open areas suitable for planting, where canopy density was less than 35%. Furthermore, a survey was conducted to investigate autumn-colored vegetation in Fuzhou City. To better guiding practice, we conducted an analysis of tree species in the surrounding mountain parks to determine which plants could be used in virtual scenarios. Eight mountain parks located in urban, suburban, and peri-urban areas of Fuzhou were examined. Ten sample plots were established in each mountain park, with a standardized sample square of 20 m × 20 m used for vegetation surveys. The typical sampling method was employed, recording the names, species, and number of trees in the sample plots. Frequency and relative frequency values were calculated, revealing that the frequency of autumn-colored plants ranged from 3.33% to 23.33%. Notably, plants displaying autumn colors accounted for more than 10% of the frequency and were primarily represented by Hackberry (*Celtis sinensis*), Maple (*Liquidambar formosana*), Ceiba (*Bombax malabaricum*), Paper Mulberry (*Broussonetia papyrifera*), Sapindus (*Sapindus mukorossi*), and Chinese Tallow Tree (*Sapium sebiferum*).

The photomontage method was employed to visualize different scenes. Following the Weber–Fechner law [16], the visualization scenes were created based on the proportion of colored vegetation elements in the scene, specifically 15%, 22.5%, 33.75%, and 50.63%, representing slight, low, medium, and high colorization design intensities, respectively. In total, 31 images were generated, including 13 canopy scenes and 18 forest scenes, all presented in a resolution of 1024 × 768 pixels (Figures 2 and 3).

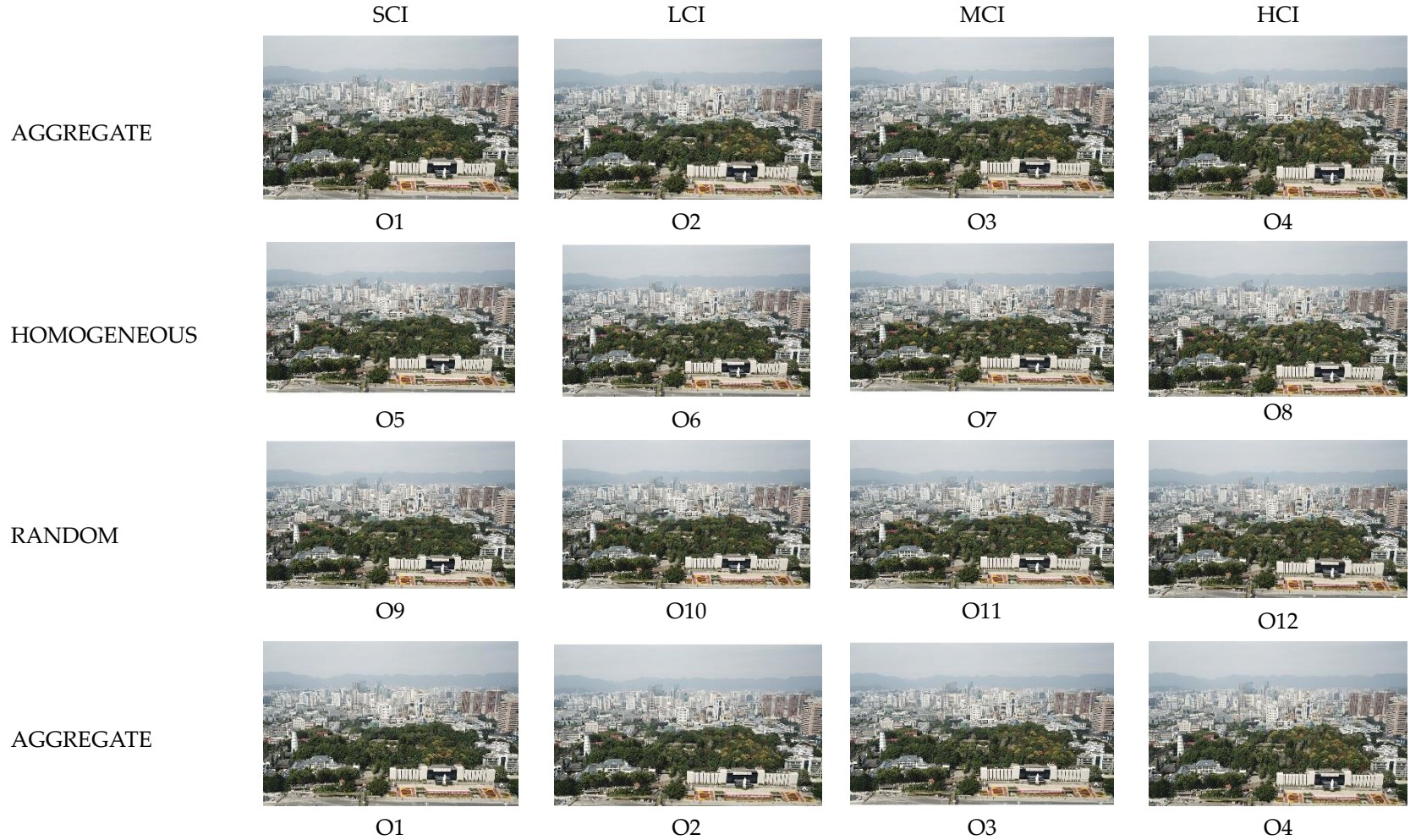

**Figure 2.** Canopy scenes were created using the photomontage technique. (SCI: slight colorization intensity, LCI: low colorization intensity, MCI: medium colorization intensity, HCI: high colorization intensity).

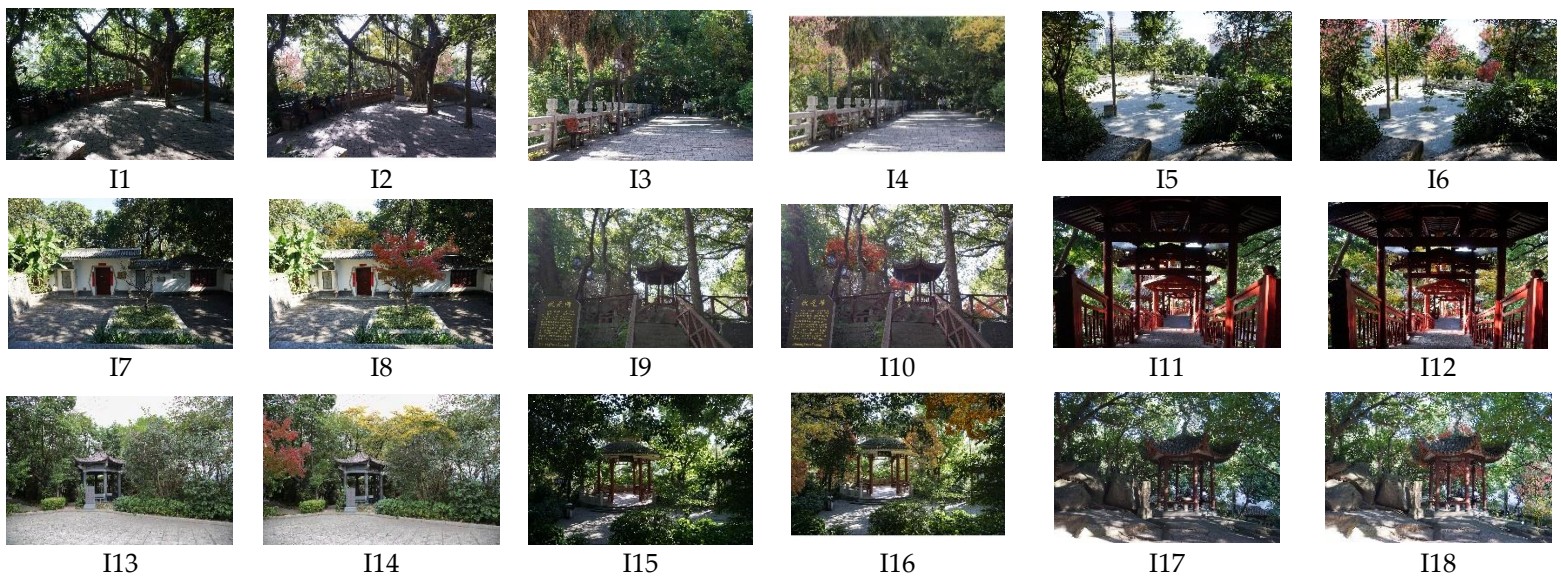

**Figure 3.** Forest scenes were created using the photomontage technique.

### 2.3. Eye-Tracking Measures

The experiment was conducted in October 2022 at the laboratory of Fujian Agriculture and Forestry University. To track eye movements, the Eyelink 1000 Plus eye tracker (SR Research Ltd., Ottawa, ON, Canada) was utilized, which has a sample rate of 1000 Hz/2000 Hz with a 2 ms interval. The eye tracker's parameters included a horizontal range of $\pm 30°$, a vertical range of $\pm 20°$, and a calibration mode of Hv9 (9 grid points). The images were presented at the center of a 19-inch screen, while a chin rest was employed to maintain a distance of 55 cm between the participant's head and the screen. Eye-movement indicators such as pupil size, fixation (including fixation count, fixation start time, and percentage of dwell time), and saccade (including saccade amplitude and saccade duration) were recorded. The fixation heatmap was generated using the Data Viewer software version 4.1.477 (SR Research Ltd., Ottawa, ON, Canada).

### 2.4. Procedure

The thirty-one images used in the experiment were randomly divided into three groups. Before the commencement of the experiment, participants were provided with a detailed overview of the experiment and instructions on how to operate the equipment. Each participant underwent the testing individually within a room illuminated by low-intensity indirect lighting, maintaining consistent light conditions throughout the recordings. Participants were seated in front of a computer screen at a distance of 55 cm, and a chin rest was utilized to stabilize their head position and ensure a constant viewing distance. They were instructed as follows: "In this experiment, you will be shown a series of autumnal photos on the computer screen. Please evaluate the aesthetic quality of the mountain landscapes using a seven-point Likert scale. Please do not try to remember or compare any details in the photos. This is not a memory test, and no tasks related to the content will appear at the end of the experiment. This experiment just wants to measure your eye movements when you look at these scenes. Before each photo presentation, a fixation point will appear on the screen". Following the methodology of previous studies [32,33,43], an observation period of 15 s was selected. Subsequently, participants rated the aesthetic value of each photo, and the experimenter recorded their responses (Figure 4).

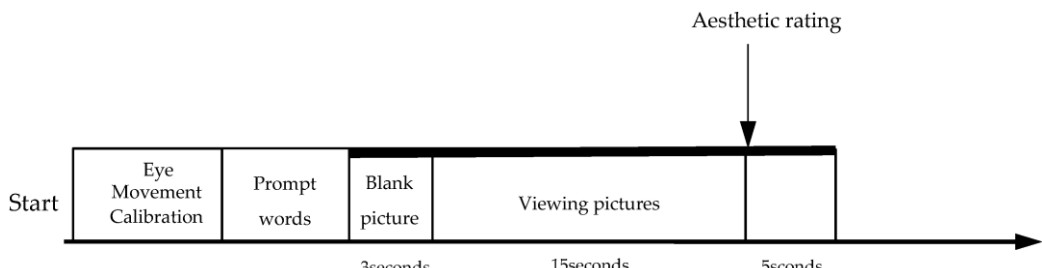

**Figure 4.** Experimental procedure.

### 2.5. Participants

Based on previous studies, the college student population is representative of the experimental study population [18]. Meanwhile, Marco demonstrated that a sample size of 30 is representative in eye-movement studies [30]. A total of 183 participants were recruited for this research from the full-time student population at the University of Agriculture and Forestry in Fujian, China. The participants consisted of 81 males and 102 females, with an average age of $23.5 \pm 2.5$ years. Before starting the experiment, each participant was subjected a vision test. All participants had normal vision or vision corrected to normal (using glasses). To ensure equal distribution, the 183 participants were divided into three groups, with each group consisting of 61 participants.

## 2.6. Image Segmentation Based on Color Statistics

In order to investigate the extent to which participants' eye movements were drawn to colored areas, image calculations were conducted. Following the approach outlined by Mirmehdi and Shang Tao [44], color segmentation and image calculations were performed within the color regions of the images. Each image was divided into 520 grids based on color threshold segmentation. These grids were then labeled as either "0" or "1" to represent uncolored and colored areas, respectively. Subsequently, an overlay analysis was carried out with the eye movement hotspots, specifically the fixation areas and their corresponding numbers (Figure 5).

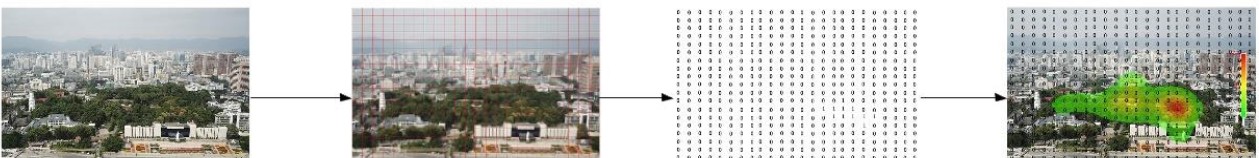

**Figure 5.** Image segmentation flow chart based on color labeling.

## 2.7. Analysis and Statistics

Eye-movement data were exported using Data Viewer software, and statistical analysis was conducted using SPSS 22.0 (IBM, Armonk, NY, USA). To ensure the reliability of aesthetic ratings, interclass reliability was tested initially. The one-way ANOVA was then employed to examine potential differences in participants' aesthetic ratings and eye movement metrics across different colorization design intensities and forms. To identify specific points of divergence, post hoc pairwise comparison tests were utilized. Furthermore, bivariate and multivariate correlation analyses were conducted to investigate the relationships between colorization design forms and intensity, aesthetic ratings, and eye-movement metrics.

## 3. Results

### 3.1. Reliability

To assess the interclass reliability of the visual aesthetic ratings, Cronbach's alpha, a statistical measure used to assess the reliability of a scale or questionnaire, was separately computed for forest scenes and canopy scenes. The Cronbach's alpha values for the forest scenes and canopy scenes were 0.929 and 0.946, respectively, indicating a high level of reliability in the questionnaire results.

### 3.2. Comparison of an Aesthetic Rating between the Original Image and Visualized Images

3.2.1. Canopy Landscapes

For canopy landscape, the mean aesthetic score of different colorization intensity scenes (the original scene was considered as zero colorization intensity) were 3.625, 4.500, 4.651, 4.910, and 4.668, respectively, on a seven-point scale (Figure 6). Notably, scenes with HCI in aggregate design form possessed a higher aesthetic value. Conversely, scenes with MCI in the random and homogeneous design form displayed higher aesthetic values. The one-way ANOVA results indicated that there was a significant difference between the five groups of colorization intensity scenes ($p < 0.01$). All the pairwise comparisons indicated no significant difference except for the original scene vs. MCI scenes ($p < 0.05$) and the original scene vs. HCI scenes ($p < 0.05$). In terms of design forms, the homogeneous scenes achieved the higher aesthetic value (4.704). The one-way ANOVA showed significant differences between the three groups of colorization design forms and the original scene ($p < 0.01$). Pairwise comparisons showed significant differences between aggregate form scenes and all the other groups ($p < 0.01$), while the rest were not different.

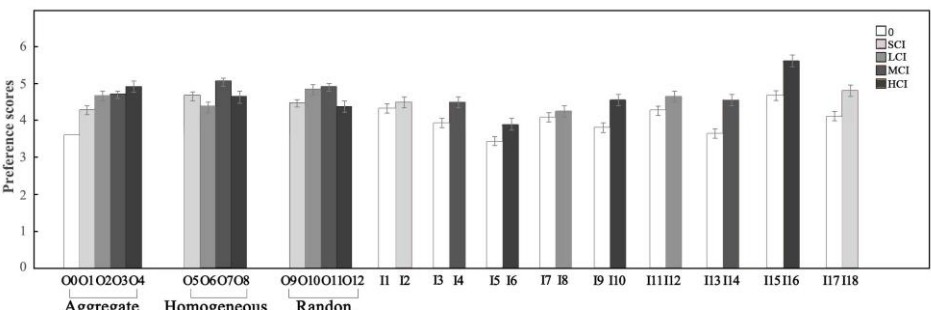

**Figure 6.** Mean aesthetic ratings for 31 images.

The marginal effects of four design intensities under three design forms on aesthetic enhancement were calculated (Figure 7). The HCI with aggregate form showed the best marginal effect (1.313). In contrast, MCI showed the best marginal effect when the design form was random and homogeneous (1.438, 1.308). In addition, according to Figure 7, the marginal benefits of MCI were more aggregated overall (SD = 0.945). The HCI scenes yielded a greater volatility of scores (SD = 1.096). Which indicates a stronger correlation between the distribution forms of MCI compared to HCI scenes.

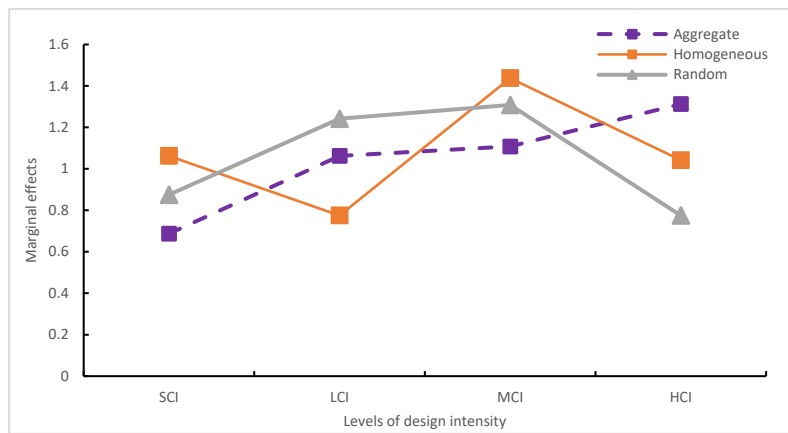

**Figure 7.** Marginal effects of four design intensities on aesthetic rating in each design form.

3.2.2. Forest Landscapes

In the case of the forest landscape, there was a gradual increase in the aesthetic value of scenes with different colorization intensities, following the pattern of the original scene < SCI < LCI < MCI < HCI. The one-way ANOVA analysis revealed significant differences among the five groups of colorization intensity scenes ($p < 0.05$). The pairwise test indicated a significant difference only between MCI and the original scenes ($p = 0.004$), while the remaining pairwise comparisons were not statistically significant. This finding suggests that the aesthetic quality of the landscape can be significantly enhanced when the colorization design intensity is at a medium level.

*3.3. Comparison of Eye Movement between the Original Image and Visualized Images*
3.3.1. Pupil Size

The results of the one-way ANOVA analysis revealed a significant difference among the five colorization intensity groups for both canopy and forest landscape scenes ($p < 0.01$). In the pairwise comparisons for canopy landscape scenes, all five colorization intensity groups showed significant differences in pupil size ($p < 0.01$). Regarding the forest landscape scenes, only the MCI and HCI scenes exhibited significant differences compared to the original scene ($p < 0.05$). Conversely, with respect to colorization design forms, the one-way ANOVA analysis indicated significant differences among the three groups

($p < 0.01$). The pairwise comparisons revealed a significant difference only between random and aggregated scenes ($p < 0.01$).

### 3.3.2. Saccade

The one-way ANOVA showed that neither the saccade amplitude nor the saccade duration showed significant differences ($p < 0.01$) in canopy landscape scenes with different colorization intensities and forms. For the forest landscape, only the saccade amplitude showed significant differences ($p < 0.01$). The further pairwise comparisons showed significant differences only for the original vs. SCI scenes ($p < 0.01$) and the original vs. HCI scenes ($p < 0.01$).

### 3.3.3. Fixation

The differences of fixation behaviors in 31 scenes were compared. The results showed that there was an inter-group difference in fixation start time of canopy scenes ($p < 0.05$) and fixation count of forest scenes ($p < 0.05$). Specifically, a significant difference was observed between the MCI and original scenes ($p < 0.05$), while no other significant differences were found. Regarding the colorization design form, variations were observed in the fixation start time and the percentage of dwell time. Pairwise analysis showed significant differences in the percentage of dwell time in all inter-group comparisons. However, for the fixation start time, the analysis indicated differences only in random vs. aggregate scenes ($p < 0.01$) (Figure 8).

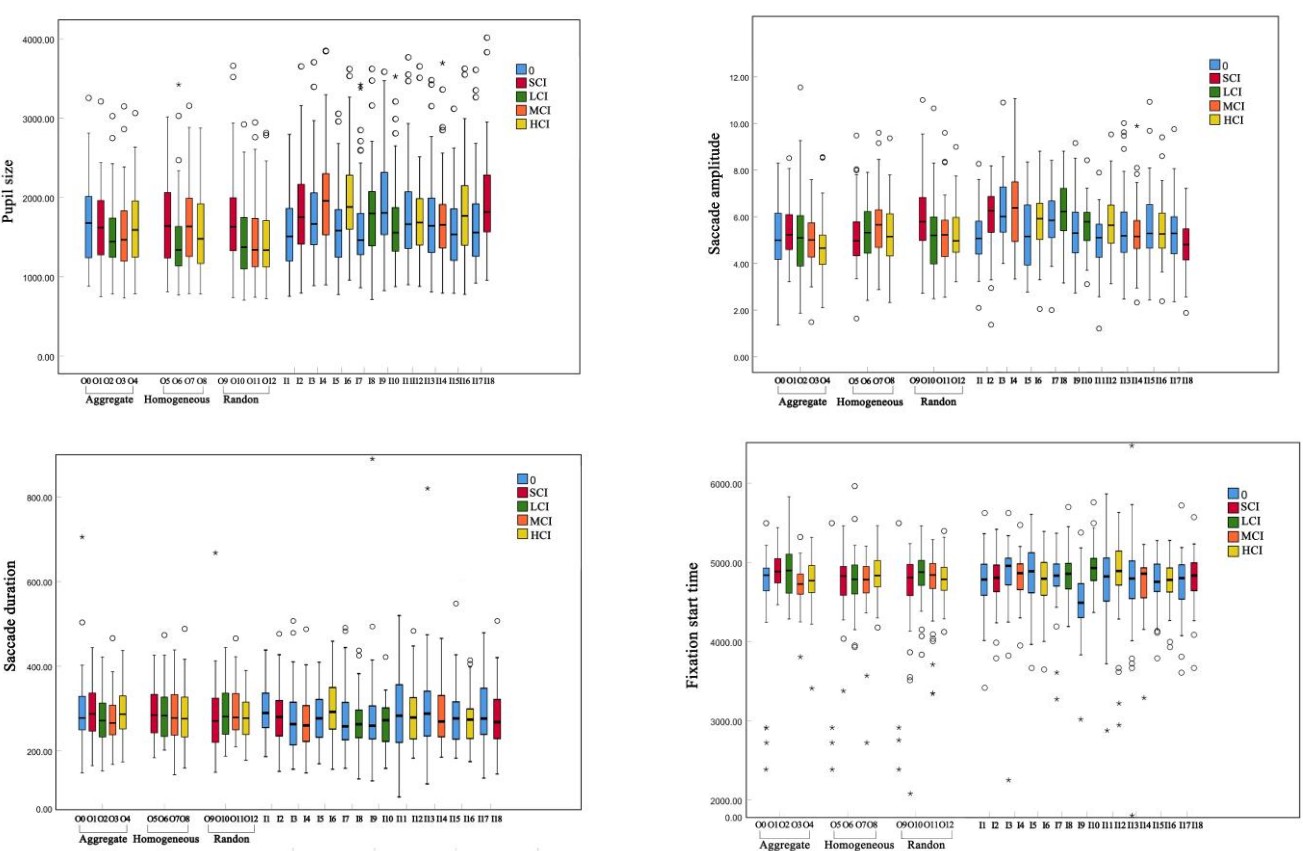

**Figure 8.** *Cont.*

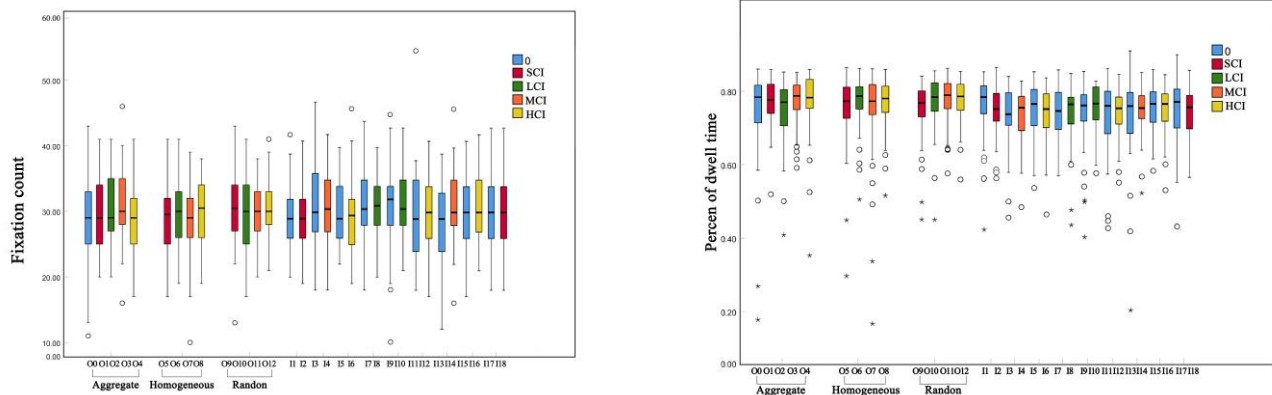

**Figure 8.** Eye movement of different scenes. Note: ○: mild outlier; *: extreme outlier.

### 3.4. Eye Movement in Relation to Colorized Areas

Pixel segmentation and superposition analysis were conducted on 30 scenes to examine eye-movement hotspots. Pearson's correlation test revealed a significant positive correlation between colorized areas and fixation metrics, including fixation area and fixation count, in the scenes ($p$ = 0.000). These findings indicate that the colorization design of urban forest scenes has a significant ability to attract objective attention (Figure 9).

|  |  |  |  |  |  |
|---|---|---|---|---|---|
| O1 | O2 | O3 | O4 | O5 | O6 |
| O7 | O8 | O9 | O10 | O11 | O12 |
| I1 | I2 | I3 | I4 | I5 | I6 |
| I7 | I8 | I9 | I10 | I11 | I12 |
| I13 | I14 | I15 | I16 | I17 | I18 |

**Figure 9.** Heatmap of different scenes.

*3.5. Eye movement in Relation to Colorization Design Form, Intensity, and Aesthetic Rating*

The colorization design form and intensity significantly influenced the aesthetic rating ($p < 0.01$), whether individually or in combination, as shown in Table 1. The correlation data were fit for correlation analyses by conforming to the normal distribution test (Figure 10). Specifically, among the eye-movement metrics, only saccade was significantly affected by the combination of form and intensity, while fixation and pupil size did not. The changes in colorization design form had a positive effect on the saccade (saccade amplitude, $p < 0.01$). On the other hand, changes in design intensity had effects on pupil size ($p < 0.05$), saccade (saccade amplitude, $p < 0.01$), and fixation (fixation count, $p < 0.01$). However, the aesthetic value was found to have a significant and negative correlation with subjective pupil size ($p < 0.01$), but no significant correlation was observed between saccade and fixation.

**Table 1.** Correlation analysis of eye movement in relation to design form, intensity, and preference ratings.

| | Eye Movement Metrics | | | | | | |
| | Pupil Size | Saccade Amplitude | Saccade Duration | Fixation Count | Fixation Start Time | Percentage of Dwell Time | Aesthetic Value |
|---|---|---|---|---|---|---|---|
| Form | −0.50 | 0.139 ** | −0.012 | 0.029 | 0.017 | 0.069 | 0.195 ** |
| Intensity | −0.90 * | −0.107 ** | −0.028 | 0.041 ** | 0.068 | 0.026 | 0.277 ** |
| Aesthetic value | −0.123 ** | −0.055 | −0.013 | −0.002 | −0.008 | −0.015 | |

Note: *: $p \leq 0.05$, **: $p \leq 0.01$.

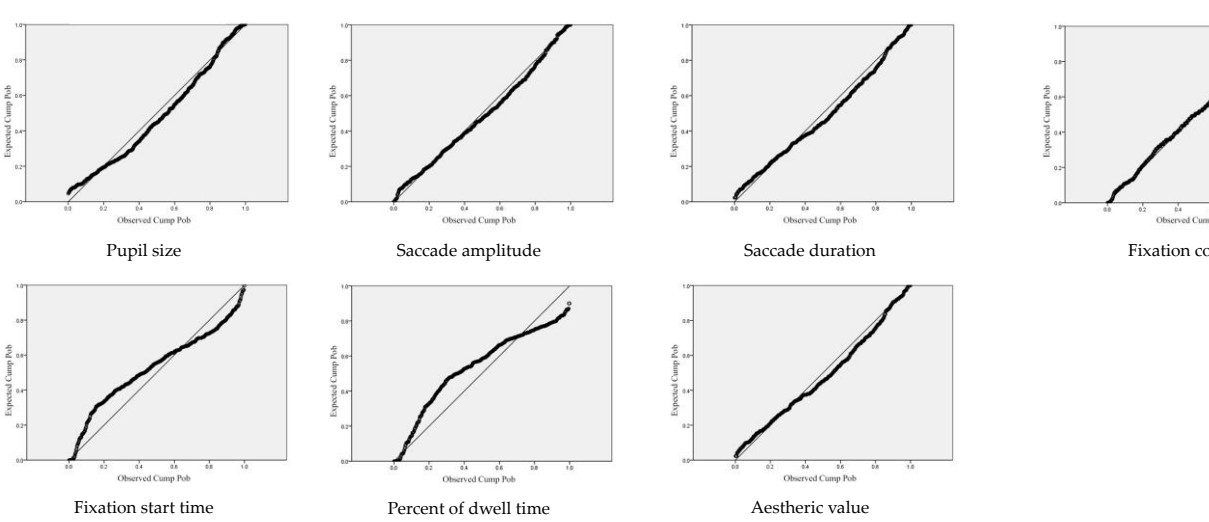

Pupil size    Saccade amplitude    Saccade duration    Fixation count

Fixation start time    Percent of dwell time    Aesheric value

**Figure 10.** Normal P–P plot of correlation data.

## 4. Discussion

*4.1. Colorization Design Form, Intensity, and Aesthetic Rating*

Previous research has indicated a preference for landscapes that incorporate natural elements with visual impact [45]. This research contributes to this body of evidence by demonstrating that scenes with vegetation colorization design exhibit higher aesthetic values compared to scenes without such design. In the context of canopy landscapes, the random and homogeneous colorization design forms combined with MCI yielded greater marginal benefits. Conversely, the aggregate colorization design form showed high marginal benefits for scenes with HCI. This may be attributed to the striking contrast between the vibrant colors of vegetation and the forest canopy background, resulting in a visually strong impact that tends to align with people's aesthetic preferences. Taking a holistic view, the MCI scenes in canopy landscapes demonstrated higher aesthetic value and more concentrated wave amplitude performance. This finding suggests that moderate design intensity is more likely to resonate with the public. In the case of forest landscapes,

the aesthetic value gradually increased as the colorization design intensity increased, with only the MCI scenes showing significant differences compared to the original scene. These findings still support the fact that scenes with MCI tend to receive higher public approval. The intermediate disturbance hypothesis, proposed by Connell (1978), emphasizes that moderate disturbance leads to minimal interference [46]. Iversen suggested that incorporating colored vegetation can induce visual changes in urban forest scenes, thereby enhancing their visual impact and subsequently affecting the aesthetic value [21,47]. Groot confirmed that, regardless of a scene type, a moderate design intensity transformation received the highest aesthetic preference [48]. The results of this research align with these findings. Furthermore, this research revealed that the homogeneous colorization design form may be more preferable compared to the random and aggregate forms. This preference can be attributed to the traditional Chinese concept of "harmony", which emphasizes homogenization and the ability to achieve an overall balance in distribution. The study results support this cultural preference. Additionally, Appleton's habitat theory (1975) suggests that landscapes with high coherence are more favored by individuals. In this research, the homogeneous design form of colored vegetation contributes to a more balanced and coherent overall scene, which is preferred by people.

### 4.2. Colorization Design Form, Intensity, and Eye Movement

This research revealed a compelling relationship between colorization design forms, intensities, and eye movements. Specifically, a significant positive correlation was observed between colored areas and fixation, indicating that the inclusion of colorful vegetation effectively attracts the viewers' gaze. This finding aligns with Huang's previous conclusion that high chromatic variation and chroma contribute to increased fixation [49]. Moreover, the research demonstrated a strong association between colorization design forms and saccade. Jiang's study emphasized that visual elements of interest within a scene can stimulate visual exploration [50]. Colorful vegetation, serving as a visual element of interest, rapidly captures viewers' visual attention, while the various distribution forms in the scene promote the occurrence of saccades. This finding is consistent with the observations of Hess and Polt (1960), who highlighted that recurring elements within a landscape significantly capture viewers' fixations [51]. As the number of elements of interest in a scene increases, so does the number of attention points, resulting in a higher fixation count. Additionally, as the design intensity increases, the aesthetic value is enhanced, subsequently influencing pupil size. Meanwhile, a new finding has been made in this research by comparing different colorization design forms. This innovative finding facilitates the enhancement of urban forest landscapes in the best way possible with limited financial and material resources.

### 4.3. Aesthetic Rating and Eye Movement

Prior investigations have established that highly appealing images resulted in less eye exploration and fewer fixation counts [52]. Wu et al. reported a significant correlation between fixation, percentage of dwell time, and aesthetic preference [53]. Landscapes with high ornamentation tend to facilitate larger pupil sizes, longer fixation durations, and reduced visual exploration. Intriguingly, the present research observed a significant correlation between aesthetic value and pupil size, while no such correlation was found with fixation and saccades. This aligns with Liu's findings, which found no direct correlation between landscape preferences for different types of green spaces and fixation [54]. Similarly, Nordh et al. reported comparable fixation counts between urban landscapes featuring varying levels of restorative elements [55]. Taken together, these outcomes suggest that saccades or fixations may not directly correlate with the aesthetic value of a landscape and cannot be employed as sole indicators thereof. Decorative elements in a landscape, characterized by intense color contrast, may attract visual attention and induce fixation and saccades [56]. However, this does not necessarily correspond to the aesthetic value of the landscape. The aesthetic judgment of a landscape constitutes a series of perceptual processes that are mediated by eye movements and subsequently reflected in the cerebral

cortex to form the final judgment. Eye movements solely serve as the initial source of sensory information, and the assessment of landscape aesthetics necessitates further information processing within the brain. In this regard, future research incorporating brain information processing is deemed necessary [57].

*4.4. Limitations*

One noteworthy limitation of this research pertains to the homogeneity of the participant group. Although the sample size in this research is statistically significant compared to existing research, it is crucial to acknowledge that aesthetic judgment is influenced by multiple factors such as culture, age, and gender [58,59]. Consequently, the generalizability of the findings may be somewhat constrained due to the common features of university students. Future investigations should consider recruiting participants from diverse backgrounds to enhance the external validity of the research. Another limitation lies in the selection of a single mountain park as the research setting. The colorization design effect on forest landscapes may be influenced by different types of recreational landscapes and various landscape components. Hence, future research should encompass a broader range of recreational landscapes to provide a more comprehensive understanding of the topic. Furthermore, the inclusion of colorful vegetation introduces a multitude of colors and shapes, and the combination of different color intensities and saturations can impact the colorization effect. It would be valuable for future studies to delve into the specific impacts of different color combinations and variations on the aesthetic perception of landscapes. Moreover, the human perception of landscapes involves intricate cognitive processes and conscious judgment in the brain. Hence, in order to better understand the human perception of landscapes, it is proposed that future exploration could incorporate research on brain processing to provide valuable insights into the underlying cognitive mechanisms.

**5. Conclusions**

This research presents a novel perspective on the perception of urban forest landscapes, offering valuable insights for designers and decision makers involved in urban green space planning. The findings underscore the significant impact of colorful vegetation within urban green spaces on capturing human attention. Moreover, the combination of design form and intensity exert a substantial influence on landscape aesthetic value. Specifically, incorporating a random form and a moderate colorization design intensity appear to be effective strategies for enhancing the aesthetic appeal of urban forest landscapes. The limitation of this research is that we have a single group of participants. In addition, color intensity and saturation factors were not addressed in this research. To better understand human perception, future research should consider including EEG technology to reveal underlying cognitive mechanisms. The current research effectively integrates subjective preference data with objective eye-tracking evidence, and provides a comprehensive understanding of the relationship between visual attention, design form, and intensity in urban green space scenes. This research not only advances our understanding of the human perception of landscapes but also highlights the potential of eye-tracking technology as a valuable tool in landscape studies. Moreover, by shedding some light on the application of eye-tracking technology in landscape studies, this work contributes new knowledge to the field of the human perception of landscape and provides fresh perspectives for future studies.

**Author Contributions:** Conceptualization, Z.C., W.F. and Y.S.; methodology, Z.C. and Z.Z.; software, Y.H.; validation, Z.C., W.F. and J.D.; formal analysis, X.Y. and J.H.; investigation, Y.L.; data curation, X.Y. and Y.H.; writing—original draft preparation, Z.C., W.F. and Y.S.; writing—review and editing, Z.C., W.F., J.H. and J.D.; visualization, Z.C.; supervision, W.F.; funding acquisition, Z.C. and Z.Z. All authors have read and agreed to the published version of the manuscript.

**Funding:** This research was funded by the Education Department of Fujian Province, grant number JAT210294, and Fujian University of Technology, grant number GY-Z20087. Green Urbanization across China and Europe: Collaborative Research on Key Technological Advances in Urban Forests, grant number 2021YFE0193200; Horizon 2020 strategic plan: CLEARING HOUSE-Collaborative Learning irResearch, Information sharing, and Governance on How Urban tree-based solutions support Sino. European urban futures, grant number 821242.

**Data Availability Statement:** No new data were created or analyzed in this study. Data sharing is not applicable to this article.

**Conflicts of Interest:** The authors declare no conflict of interest.

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
