# Peer review of "How Vegetation Colorization Design Affects Urban Forest Aesthetic Preference and Visual Attention: An Eye-Tracking Study"

_forests, doi:10.3390/f14071491_

Round 1

Reviewer 1 Report

Dear authors, 

There are my comments.

1.  Introduction part: good.

2. The scale is missing in Figure 1.

3. Each observasion site should be marked in Figure 1 or other Figure.

4. The visual condition of the participants should be mentioned. Do these persons include short-sighted? the naked eyes?

5.Why are the participants all young?" 23.5 ± 2.5 years"

6. You used Pearson's correlation, but there is no mention of the normal distribution test.

7. Discussion part: 4.2 should be more discussed.

8. Conclusions part: good.

Best regards.

Author Response

Response to Reviewer 1 Comments

Comments and Suggestions for Authors

Dear authors, 

There are my comments.

  1. Introduction part: good.
  2. The scale is missing in Figure 1.

Response 2:Thank you so much for this suggestion. The scale has been added in revised manuscript (Figure 1).

  1. Each observasion site should be marked in Figure 1 or other Figure.3.

Response 3:Thank you so much for this suggestion. Each observasion site have been added in Figure 1.

  1. The visual condition of the participants should be mentioned. Do these persons include short-sighted? the naked eyes?

Response 4:Thank you so much for this suggestion and sorry for the unclear expression. We have revised the section 2.5.

5.Why are the participants all young?" 23.5 ± 2.5 years"

Response 5:Thank you for your efficient work and the question. Previous researches have verified the representativeness of university student population as experimental research subjects. Therefore, this research recruited 183 college students as participants. It also leads to younger participants. We have revised the section 2.5.

  1. You used Pearson's correlation, but there is no mention of the normal distribution test.

Response 6:Thank you so much for this suggestion. The normal distribution test has been added in the manuscript (Result part:3.5).

  1. Discussion part: 4.2 should be more discussed.

Response 7:Thank you so much for this suggestion. The 4.2 of discussion part has been revised in manuscript (Discussion part: 4.2).  

  1. Conclusions part: good

Reviewer 2 Report

The author submitted a manuscript that is interesting, and it is linked to the objectives of the journal, however, there are some issues that have to be reconsidered.

The objective of the manuscript is to provide original insight into the impact of colorization design on aesthetic benefits and eye movements in urban forests, considering four intensities (slight, low, medium, and high) and three forms (aggregate, homogeneous and random).

The subject area is rather interesting, and, possibly, not enough approached by other scholars, so there is potential room for this manuscript to bring new information, once it reaches the expected level of quality.  

The Abstract has to be reconsidered, providing more information about the representative results (mainly the measurable ones).

In the introduction, the objective of the study is detailed and defined, but the structure of the manuscript should be mentioned. The part of the Literature Review is part of the Introduction, and looks a bit superficial, so the literature gap could be better pointed out.

The methodology part. It is necessary to explain why the selected 183 experts are representative of the entire studied population.

The Results. The results are interesting and well-constructed and, also, The discussion. Part is consistent.

The conclusion. Limits of the research must be mentioned.

Author Response

The author submitted a manuscript that is interesting, and it is linked to the objectives of the journal, however, there are some issues that have to be reconsidered. The objective of the manuscript is to provide original insight into the impact of colorization design on aesthetic benefits and eye movements in urban forests, considering four intensities (slight, low, medium, and high) and three forms (aggregate, homogeneous and random). The subject area is rather interesting, and, possibly, not enough approached by other scholars, so there is potential room for this manuscript to bring new information, once it reaches the expected level of quality.  

  1. The Abstracthas to be reconsidered, providing more information about the representative results (mainly the measurable ones).

Response 1:Thank you so much for this suggestion. More information about the representative results have been added in the abstract of revised manuscript (line 24, 27-30).

  1. In the introduction, the objective of the study is detailed and defined, but the structure of the manuscript should be mentioned. The part of the Literature Reviewis part of the Introduction, and looks a bit superficial, so the literature gap could be better pointed out.

Response 2:Thank you so much for this suggestion. The structure of the manuscript have be mentioned in manuscript (line 115-119). And the literature gap has been pointed out in line 111-115.

  1. The methodology part. It is necessary to explain why the selected 183 experts are representative of the entire studied population.

Response 3:Thank you so much for this suggestion. Previous researches have verified the representativeness of university student population as experimental research subjects. Therefore, this research recruited 183 college students as participants. It has been explained in the manuscript (line 216-217). Meanwhile, Marco (Marco, 2018) demonstrated that a sample size of 30 is representative in eye movement studies, and in this research we chose a sample size of 183.

  1. The Results. The results are interesting and well-constructed and, also,The discussion. Part is consistent. The conclusion. Limits of the research must be mentioned.

Response 4:Thank you so much for this suggestion. Limits of the research have been added in the manuscript (conclusion part).

Reviewer 3 Report

I find the article very interesting, eye-tracking research is increasingly the focus of attention of landscape architects, slowly also foresters. I have a few minor comments on the whole work. in the „Introduction” section - it is worth adding a thread about the practical importance of landscape assessment. Landscape preferences are, after all, of great importance in real estate purchase/sale transactions. It has long been known that marketing professionals make use of landscape preference findings in order to achieve, for example, greater sales efficiency. Eye-tracking research is important for improving road user safety. Therefore, it seems to me that it is worth highlighting the practical side of the research undertaken. materials and methods - I wonder about the thread described in section 2.2. line 149-162. In fact, in the rest of the paper in the results there is no direct reference to the details discussed. Therefore, it seems to me that this element is too much developed. participants - how do the authors of the study have information and confidence that the participants had "normal vision." What does this mean, is it just a visual defect consisting of short/far sightedness (using glasses), or is it also daltonism? How was this taken into account in the study? Very aptly stated limitations. Indeed, the survey was conducted only on a sample of young people, so I think this theme should be more strongly articulated in the title of the paper. I suggest adding - that it is about the preferences of young people

Author Response

  1. in the „Introduction” section - it is worth adding a thread about the practical importance of landscape assessment. Landscape preferences are, after all, of great importance in real estate purchase/sale transactions. It has long been known that marketing professionals make use of landscape preference findings in order to achieve, for example, greater sales efficiency. Eye-tracking research is important for improving road user safety. Therefore, it seems to me that it is worth highlighting the practical side of the research undertaken.

Response 1:Thank you so much for this suggestion, we have revised in Line 111-120.

  1. materials and methods - I wonder about the thread described in section 2.2. line 149-162. In fact, in the rest of the paper in the results there is no direct reference to the details discussed. Therefore, it seems to me that this element is too much developed.

Response 2:Thank you so much for this suggestion. Section 2.2 line 152-160 was described in order for readers better understand the realistic basis of how the experiment was conducted. According to your suggestions, we have rewritten section2.2 to make it more understandable.

  1. participants - how do the authors of the study have information and confidence that the participants had "normal vision." What does this mean, is it just a visual defect consisting of short/far sightedness (using glasses), or is it also daltonism? How was this taken into account in the study?

Response 3:Thank you so much for this suggestion, and sorry for the unclear expression. The “vision corrected to normal” is just a visual defect consisting of short/far sightedness (using glasses). All participants were tested before participating in the experiment to see if they had normal vision (including corrected vision). Participants with daltonism or other visual deficiencies were excluded from the experiment. According to your suggestions, we have rewritten section 2.5 to make it more understandable.

  1. Very aptly stated limitations. Indeed, the survey was conducted only on a sample of young people, so I think this theme should be more strongly articulated in the title of the paper. I suggest adding - that it is about the preferences of young people

Response 4:Thank you so much for this suggestion. We have added this restriction to the abstract of the article. And we have added a note in section 2.5.
